# Keratinocyte interleukin-36 receptor expression orchestrates psoriasiform inflammation in mice

Yasmina E Hernández-Santana[1,2] , Gemma Leon[1,2] , David St Leger[1,2], Padraic G Fallon[1], Patrick T Walsh[1,2]

The IL-36 family cytokines have emerged as important mediators of dermal inflammation in psoriasis and have been reported to provide a proinflammatory stimulus to a variety of immune and stromal cell subsets in the inflamed skin. However, it remains to be determined which cell type, if any, in the skin plays a predominant role in mediating IL-36 cytokines instructive role in disease. Here, we demonstrate that targeted deletion of *Il36r* in keratinocytes results in similar levels of protection from psoriasiform inflammation observed in "global" *Il36r*-deficient mice. Mice with deficiency in IL-36 receptor expression on keratinocytes had significantly decreased expression, comparable with *Il36r*-deficient mice, of established mediators of psoriatic inflammation, including, IL-17a, IL-23, IL-22, and a loss of chemokine-induced neutrophil and IL-17A–expressing γδ T-cell subset infiltration to the inflamed skin. These data demonstrate that keratinocytes are the primary orchestrating cell in mediating the effects of IL-36–driven dermal inflammation in the imiquimod model of psoriasiform inflammation and shed new light on the cell-specific roles of IL-36 cytokines during psoriatic disease.

## Introduction

Psoriasis is a common debilitating autoinflammatory disease of the skin which represents a massive socioeconomic burden, affecting up to 3% of the population, with profound impacts on patient's physical and psychological well-being (Gelfand et al, 2004; Mrowietz et al, 2014). The IL-36 family of cytokines is emerging as central orchestrating mediators of psoriatic disease. The family consists of three separate agonistic ligands, designated IL-36α, IL-36β, and IL-36γ and a specific IL-36 receptor antagonist all of which act through a specific IL-36 receptor (Walsh & Fallon, 2018). Similar to the more extensively characterised "classical" IL-1 cytokines, IL-1α and IL-1β, IL-36 cytokines are also thought to act as important drivers of inflammation but function in a more tissue restricted manner (Gabay & Towne, 2015). A particular focus on the role of IL-36 family members as orchestrators of inflammatory skin disease has

recently emerged. This focus stems from the identification of specific mutations in the gene encoding the IL-36 receptor antagonist (*IL36RN*), which reduce its activity leading to the development of an autoinflammatory condition characterised by a severe form of generalized pustular psoriasis (Marrakchi et al, 2011; Onoufriadis et al, 2011). Numerous studies, using both murine models of psoriasis, as well as patient tissues, have expanded and effectively translated these discoveries, further implicating IL-36 family members as central orchestrators of dermal inflammation, even in more prevalent forms of disease such as psoriasis vulgaris (Blumberg et al, 2007, 2010; Carrier et al, 2011; Tortola et al, 2012; Bachelez et al, 2019). This has prompted a significant effort among pharmaceutical companies to develop monoclonal antibody strategies aimed at targeting the IL-36R signaling axis for the treatment of psoriasis, which have very recently been validated in early clinical trials (Ganesan et al, 2017; Mahil et al, 2017). Although the central role of IL-36 cytokines in psoriatic inflammation has been established, many questions surrounding their precise mechanism of action in this setting remain unanswered. IL-36 receptor stimulation has been reported to promote proinflammatory responses in various skin cell subsets, including keratinocytes, fibroblasts, macrophages, dendritic cells, and various T cell subsets (Towne & Sims, 2012; Walsh & Fallon, 2018). However, the question of which responding cell types, if any, are critical in driving IL-36–dependent dermal inflammation is unclear.

We have addressed this question using a newly generated transgenic mouse (*Il36rΔK*), demonstrating that keratinocyte-specific expression of Il36r is required to orchestrate psoriasiform disease. Critically, Il36r deficiency in keratinocytes only mirrored the protection from psoriasiform inflammation, induced through the topical administration of Aldara cream, observed in "globally" Il36r-deficient mice. This protection from skin inflammation occurred in association with a failure to induce the expression of established mediators of psoriatic inflammation including IL-17a, IL-23, and IL-22. In addition, although the heightened expression of IL-36 family cytokines themselves was only marginally altered in the inflamed skin of *Il36rΔK* mice, the induced expression of known IL-36–responsive genes encoding Il17c, and the antimicrobial peptides S100a8 and S100a9 were also diminished. Significantly, loss of Il36r expression in keratinocytes also resulted in a loss of infiltration of neutrophils and IL-17a–expressing Vγ4+ γδ T cells to the inflamed

[1]Department of Clinical Medicine, Trinity Translational Medicine Institute, Trinity College Dublin, Dublin, Ireland    [2]National Children's Research Centre, Dublin, Ireland

Correspondence: walshp10@tcd.ie

skin. This study demonstrates the central orchestrating role for keratinocyte-specific IL-36 responses in driving psoriasiform inflammation.

# Results and Discussion

### Loss of expression of Il36r in keratinocytes results in similar levels of protection from psoriasiform inflammation to those observed in $Il36r^{-/-}$ mice

In an effort to identify which cell types play an instructive role in mediating IL-36–driven dermal inflammation, we generated a novel *Il36r* floxed ($Il36r^{fl/fl}$) mouse to facilitate cell-specific deletion of *Il36r* gene, its expression, and responses (Fig S1). Although IL-36 family cytokines have been reported to stimulate various cell subsets of immune and stromal origin in the skin, we sought to specifically examine the role of keratinocytes given their reported expression of the IL-36 receptor among human patients, responses to IL-36 stimulation ex vivo, and established role in the pathogenesis of psoriatic disease (Blumberg et al, 2007, 2010; Carrier et al, 2011; Tortola et al, 2012; Mahil et al, 2016; Madonna et al, 2019). To address this question, we crossed the $Il36r^{fl/fl}$ mouse with $K14Cre^+$ transgenic mice, in which the Cre recombinase is expressed in keratinocytes under the control of the *keratin-14* gene promoter, to generate mice in which Il36r expression was specifically deleted among keratinocytes in the skin (*Il36rΔK* mice) (Wang et al, 1997; Dassule et al, 2000). *Il36rΔK* mice were overtly normal and showed no evidence of baseline-altered skin homeostasis or inflammation, which was comparable with that observed in $Il36r^{fl/fl}$ littermates (Fig S2). Specific *Il36r* deletion was confirmed through analysis of IL-36r protein expression in both uninflamed and inflamed skin induced through daily topical administration of 5% Aldara cream, which contains the TLR7 agonist imiquimod, for 6 d, by immunohistochemistry (Fig 1A). These data demonstrate that epidermal keratinocytes represent the major cell type expressing the IL-36r in the skin of wild-type mice and confirm that this expression is lost in *Il36rΔK* mice. We also examined the levels of gene expression of the Il36r in the inflamed skin of these mice, demonstrating that overall Il36r expression is significantly decreased in *Il36rΔK* skin (Fig 1B). Together, these data demonstrate that the Il36r is predominantly expressed in keratinocytes in inflamed skin, and this expression is lost in the *Il36rΔK* mice.

To confirm a specific loss of responsiveness to IL-36 ligands in keratinocytes, we next examined the induction of Il17c gene expression in isolated primary keratinocytes stimulated for 24 h with IL-36α. Expression of Il17c has previously been found to be up-regulated in keratinocytes in response to IL-36 stimulation (Hashiguchi et al, 2018). As predicted, IL-36 responsiveness is lost in keratinocytes isolated from *Il36rΔK* mice (Fig 1C). Importantly, and in contrast to keratinocytes, IL-36α–induced expression of Cxcl1 in *Il36rΔK* dermal fibroblasts is maintained, indicating that functional expression of the IL-36 receptor in these skin-resident stromal cells is maintained in *Il36rΔK* mice (Fig 1D). Similarly, BMDCs from *Il36rΔK* respond to IL-36α to a similar degree as BMDCs from $Il36r^{fl/fl}$ mice (Fig S3), demonstrating that functional IL-36r responses are preserved in non-keratinocytes in *Il36rΔK* mice.

To investigate the functional importance of keratinocyte-specific expression of IL-36r in the context of psoriatic inflammation, we analysed disease pathogenesis among *Il36rΔK* mice using the Aldara-induced model of psoriasiform inflammation. This model has previously been reported to be sensitive to treatment with frontline current therapeutic approaches currently in use in psoriasis patients (Pantelyushin et al, 2012; Mitsui et al, 2015; Shibata et al, 2015; Shimizu et al, 2019). Strikingly, deficiency of the IL-36r in keratinocytes alone resulted in a similar level of protection from disease to that observed in $Il36r^{-/-}$ mice, whereas wild-type $Il36r^{fl/fl}$ littermate controls developed marked disease. Furthermore, this protection was evident in terms of ear thickening (Fig 1E), as well as overall histological scoring incorporating levels of acanthosis, desquamation, parakeratosis, and infiltration, evaluated through hematoxylin and eosin staining of treated skin (Fig 1F and G).

Aldara-induced psoriasiform inflammation in mice has previously been shown to be profoundly regulated by IL-36 family cytokine signaling. In agreement with our study, mice deficient in the *Il36r* across all tissues and cell types have been shown to be largely protected from disease pathogenesis, whereas mice deficient in the *Il36rn* gene ($Il36rn^{-/-}$), encoding the IL-36 receptor antagonist, exhibited an exacerbated disease phenotype (Tortola et al, 2012). Similarly, treatment with anti-IL-36 receptor blocking antibodies can effectively suppress psoriasiform inflammation in this model (Ganesan et al, 2017; Mahil et al, 2017). Although no studies to date have investigated which specific responding cell type, if any, plays an important role in mediating these effects, Tortola et al (2012) have reported, through bone marrow chimera studies, that radio-resistant cells mediate the pathogenic effects of IL-36r signaling in the Aldara model (Tortola et al, 2012). Furthermore, relative expression levels of the IL-36 receptor were found to be higher in keratinocytes, fibroblasts, and neutrophils compared with dendritic cells and T-cell subsets (Mahil et al, 2017). Such observations are consistent with our findings that IL-36R expression on keratinocytes is required to mediate skin disease.

### Expression of Il36r on keratinocytes is required for the activation of key pathogenic pathways in psoriasiform inflammation

As a first step in evaluating the mechanism through which IL-36–dependent keratinocyte responses mediate dermal inflammation, we examined the levels of expression of several genes previously characterised as playing important roles in driving psoriatic skin inflammation. As expected, topical Aldara treatment of wild-type $Il36r^{fl/fl}$ skin induced the expression of key proinflammatory mediators such as Il17a, Il23, and Il22. However, these effects were lost in $Il36r^{-/-}$ mice and also in mice lacking expression of *Il36r* only in keratinocytes (*Il36rΔK*) (Fig 2A). In addition, the expression levels of Il17c and genes encoding the antimicrobial peptides, S100a8 and S100a9, which are induced by IL-36 cytokines specifically in keratinocytes and implicated in disease progression (Mahil et al, 2017), were also reduced in the skin of *Il36rΔK* mice (Fig 2B).

It has been previously reported that IL-36 family cytokines can promote their own expression in inflamed skin and possibly act in a feed forward fashion, in tandem with other pathogenic cytokines, to perpetuate inflammation and disease pathogenesis (Milora et al, 2015; Hernandez-Santana et al, 2019; Madonna et al, 2019). Therefore, we

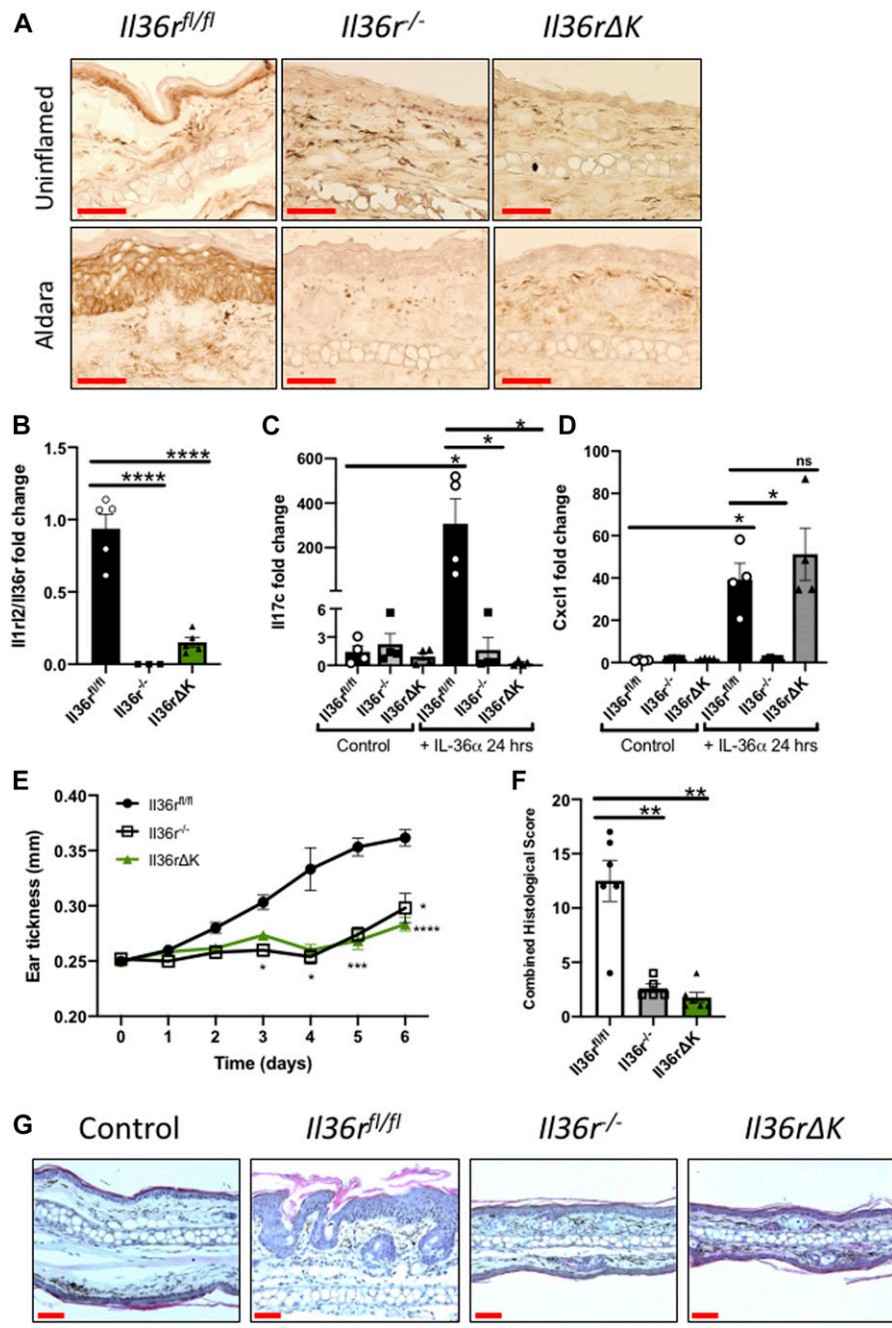

**Figure 1. Deletion of *Il36r* gene in keratinocytes results in similar levels of protection from psoriatic inflammation to those observed in *Il36r*$^{-/-}$ mice.**
**(A)** Representative micrographs obtained after immunohistological detection of IL-36R staining using DAB (brown) in fresh frozen sections from mice ears of control, *Il36r*$^{fl/fl}$, *Il36r*$^{-/-}$, and *Il36rΔK* mice after vehicle (uninflamed) or Aldara cream (5% Imiquimod [IMQ]) topical administration for 6 d. Scale bar = 1 μm. **(B)** Relative Il36r gene expression levels in the skin of *Il36r*$^{fl/fl}$ (*n* = 5), *Il36r*$^{-/-}$ (*n* = 3), and *Il36rΔK* (*n* = 5) mice after 7-d Aldara treatment. **(C, D)** Il17c gene expression levels in keratinocytes and (D) Cxcl1 gene expression levels in fibroblasts, untreated, and treated with recombinant mouse IL-36 α for 24 h. **(E, F)** Ear thickness and (F) combined histological scoring of *Il36r*$^{fl/fl}$ (*n* = 6), *Il36r*$^{-/-}$ (*n* = 5), and *Il36rΔK* (*n* = 6) mice after six consecutive days of Aldara cream (5% IMQ) topical administration. **(G)** Representative micrographs obtained after hematoxylin and eosin staining of ear sections of control (vehicle-treated Il36r$^{fl/fl}$) and *Il36r*$^{fl/fl}$, *Il36r*$^{-/-}$, and *Il36rΔK* mice after 6-d Aldara cream topical administration. Scale bar = 1 μm. Data shown in (E) are representative of three independent experiments with similar results. **(B, C, D)** Data show means ± SEM. Statistical analyses were performed using one-way ANOVA multiple comparisons test with Tukey's correction in Fig 1B–D, two-way ANOVA multiple comparisons test with Bonferroni correction in Fig 1E, and Mann–Whitney test in Fig 1F. Significant differences are indicated as follows: *P < 0.05; **P < 0.01; ***P < 0.001, ****P < 0.0001.
Source data are available for this figure.

investigated whether loss of IL-36r activity among keratinocytes might alter the induction of IL-36 family gene expression in inflamed skin. Interestingly, the induction of Il36a, Il36b, and Il36g gene expression upon Aldara treatment were largely maintained, albeit at slightly reduced levels, irrespective of *Il36r* expression (Fig 2C), indicating that factors, other than the IL-36 cytokines themselves, can drive their own expression in inflamed skin.

These data demonstrate that keratinocyte responses to IL-36 signaling are sufficient to orchestrate the activation of key pathogenic pathways in psoriasis-like dermal inflammation. Although it has previously been established that IL-23 and IL-17a expression and activity are positively regulated by IL-36 cytokines, in the context of

dermal inflammation, our findings provide an important advance in identifying keratinocytes as the key cells in mediating these effects (Tortola et al, 2012; Pfaff et al, 2017). In agreement with several previous studies, we have also found that IL-36 cytokines can induce the expression of genes expressed by keratinocytes such as Il17c and S1008a and S1009a. Our analysis of *Il36* family gene expression has revealed that IL-36 cytokines themselves play a relatively minor role in positively regulating their own expression in inflamed skin. In this regard, it is noteworthy that dendritic cells have been shown to play a critical role in mediating IL-36–dependent psoriasiform inflammation in this model. It is tempting to speculate that these cells, upon activation by mediators other than IL-36, can

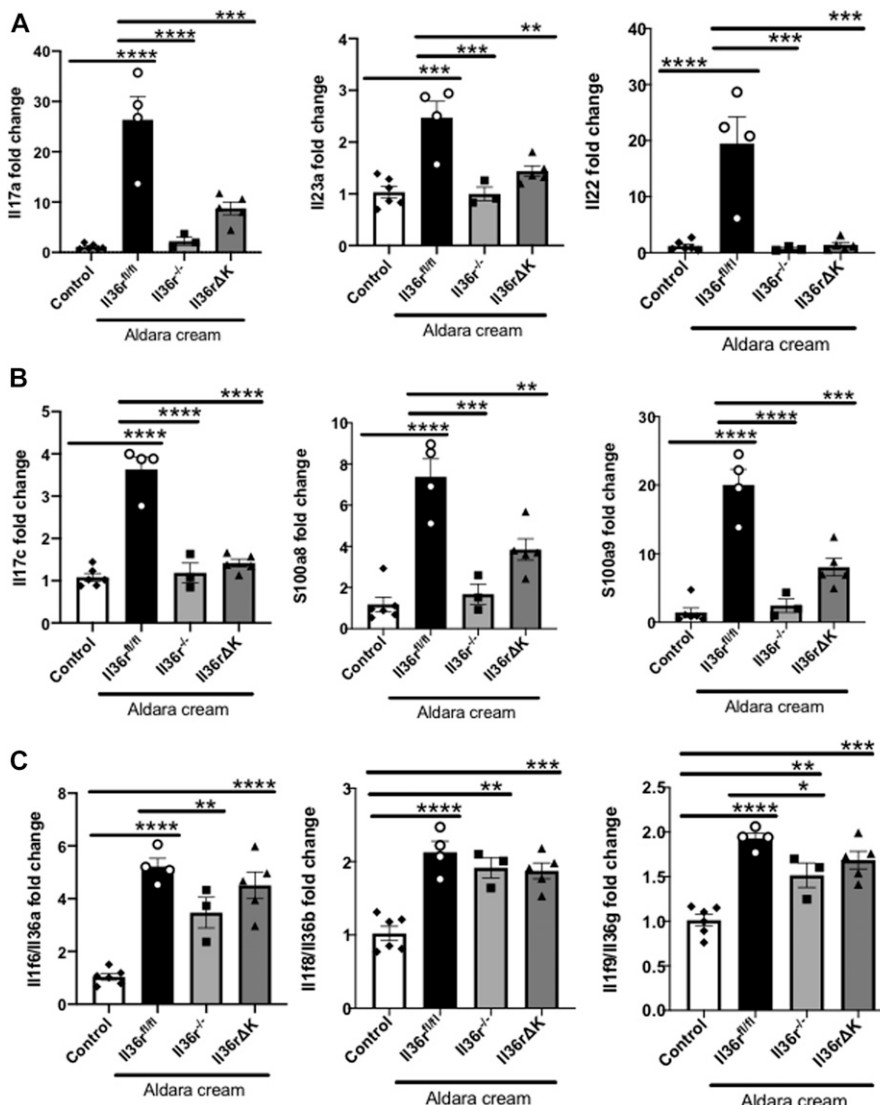

**Figure 2. Levels of induced inflammatory gene expression in the skin of *Il36r*^fl/fl^, *Il36r*^−/−^, and *Il36rΔK* mice after 4-d Aldara treatment as determined by qRT-PCR.**
**(A)** Relative gene expression levels of inflammatory mediators associated with psoriatic inflammation Il17a, Il23, and Il22 in cDNA from ear extracts of control uninflamed (*n* = 6) and *Il36r*^fl/fl^ (*n* = 4), *Il36r*^−/−^ (*n* = 3), and *Il36rΔK* (*n* = 5) mice after 4 d Aldara treatment. **(B, C)** Relative gene expression levels of keratinocyte-associated genes Il17c, S100a8, and S100a9 and, (C) relative gene expression of IL-36 family ligands, Il36a, Il36b, and Il36g. Data show means ± SEM. Statistical analysis was performed of changes in expression relative to control uninflamed skin using one-way ANOVA Multiple comparisons with Tukey's correction. Significant differences are indicated as follows: *\*P <
0.05; \*\*P < 0.01; \*\*\*P < 0.001, \*\*\*\*P < 0.0001.*
Source data are available for this figure.

provide an important source of IL-36 family expression in this setting (Tortola et al, 2012; Numata et al, 2018; Madonna et al, 2019). Together, these findings also have direct relevance to the "feed-forward" model of psoriatic inflammation, wherein T-cell–derived IL-17 family cytokines can act in synergy with IL-36 family members, as well as other inflammatory mediators, to promote keratinocyte hyperplasia characteristic of disease (Milora et al, 2015; Campbell et al, 2017; Pfaff et al, 2017). Our results indicate that IL-36–derived signaling in keratinocytes also plays an earlier role in the initiation of this amplified response through directly regulating IL-17a expression, possibly through the induction of IL-23.

### Expression of Il36r on keratinocytes promotes the recruitment of neutrophils and IL-17a–expressing γδ T cells to the inflamed dermis

Psoriasiform inflammation as a result of topical Aldara administration results in the activation and recruitment of inflammatory immune cells which contribute directly to epidermal hyperplasia.

Key cell subsets involved in this process include neutrophils and γδ T cells, which provide a major source of pathogenic IL-17a expression (Cai et al, 2011). It has previously been demonstrated that targeting IL-36r signaling in this model can suppress these responses (Tortola et al, 2012). Therefore, we next sought to examine what influence, if any, Il36r expression on keratinocytes plays in the recruitment and activation of these cells. As previously reported, Aldara treatment for 4 d led to a significant infiltration of cells of hemopoietic origin to the skin of wild-type mice when compared with baseline numbers of CD45⁺ cells present in (control) treated wild-type *Il36r*^fl/fl^ skin (Fig 3A) (Tortola et al, 2012). In contrast, Aldara treatment of *Il36rΔK*, as well as *Il36r*^−/−^, mice did not induce significant infiltration of CD45⁺ cells, demonstrating the IL-36 cytokine activity in keratinocytes is required to regulate the inflammatory cell infiltration in psoriatic skin (Fig 3A).

In terms of specific pathogenic immune cell subsets, the Vγ4⁺ subset of γδ T cells are a prominent source of IL-17a in mouse skin and are thought to clonally expand and play a central pathogenic

**Figure 3. Keratinocyte expression of Il36r directs IL-17a–producing Vγ4⁺ γδ T cells to the inflamed dermis.**

**(A)** Total numbers of CD45⁺ cells after 4-d Aldara treatment. **(B, C)** Percentage and (C) total cell counts of Vγ4⁺ γδ T cells in the ear skin of vehicle-treated (control) (*n* = 6) and Aldara-treated *Il36r^fl/fl* (*n* = 7), *Il36r^−/−* (*n* = 7), and *Il36rΔK* (*n* = 7) mice as determined by flow cytometry. **(D, E)** Percentage and (E) total cell counts of IL-17a–producing Vγ4⁺ γδ T cells in the same mice. **(F)** Representative plots of Il-17a expression levels in Vγ4⁺ γδ T cells as determined by flow cytometry. **(G, H)** Expression levels of (G) IL-17a and (H) IL-23 protein expressed as pg/mg of total protein in the skin of littermate control (*n* = 6), and *Il36r^fl/fl* (*n* = 5), *Il36r^−/−* (*n* = 3), and *Il36rΔK* (*n* = 4) mice. **(I)** Relative Ccl20 gene expression levels in the ear skin of control (*n* = 6), and Aldara-treated *Il36r^fl/fl* (*n* = 4), *Il36r^−/−* (*n* = 3) and *Il36rΔK* (*n* = 5) mice. Data show means ± SEM. Statistical

role in the inflamed dermis in the Aldara model (Hartwig et al, 2015; Prinz & Sandrock, 2015; Ramírez-Valle et al, 2015). Therefore, we examined the relative levels of Vγ4$^+$ γδ T cells, as well as their expression of IL-17a, in the inflamed skin of *Il36rΔK, Il36r$^{-/-}$*, and *Il36r$^{fl/fl}$* mice, compared with levels found in uninflamed skin. Although the percentage of Vγ4$^+$γδTCR$^+$ T cells within the CD45$^+$ compartment did not change between groups, the overall numbers of infiltrating Vγ4$^+$γδTCR$^+$ T cells were significantly elevated in the inflamed skin of *Il36r$^{fl/fl}$* control mice. In contrast, numbers of this subset were significantly, and similarly, reduced in *Il36rΔK* and *Il36r$^{-/-}$* inflamed skin (Fig 3B and C). As expected, the percentage and number of Vγ4$^+$γδTCR$^+$ T cells expressing IL-17A were also significantly increased in the inflamed skin of *Il36r$^{fl/fl}$* mice. This increase, both in terms of expression on a per cell basis, and overall cell numbers, was lost in the inflamed skin of both *Il36rΔK* and *Il36r$^{-/-}$* mice, demonstrating that keratinocyte-specific IL-36r signaling is required for the activation, recruitment, and/or expansion of IL-17$^+$Vγ4$^+$ γδ T cells in the inflamed dermis (Fig 3D–F). Importantly, this overall decrease in IL-17a–expressing cells was reflected in the amount of total IL-17a protein detected in the inflamed skin at this time point (Fig 3G), as well as a failure to induce expression of both IL-23 protein and the Ccr6 ligand, Ccl20, which is a key signal in recruiting IL-17a–expressing cells to inflamed skin (Mabuchi et al, 2011; Campbell et al, 2017) (Fig 3H and I).

In addition to IL-17a$^+$ γδ T cells, we also examined the levels of infiltration of neutrophils as a key pathogenic cell subset, previously reported to be recruited to the inflamed skin by IL-36 cytokines in this model (Tortola et al, 2012). This analysis demonstrated that the recruitment of CD45$^+$CD11b$^+$Ly6G$^+$ neutrophils was significantly diminished, as determined by percentage and overall cell number, in the inflamed skin of both *Il36rΔK* and *Il36r$^{-/-}$* mice (Fig 4A–C). This lack of neutrophil infiltration occurred in association with a failure to induce protein expression of the neutrophil chemo attractant Cxcl1 in the inflamed skin. Cxcl2 levels were also examined and found to be somewhat reduced in *Il36rΔK* skin, albeit not to significant levels (Fig 4D and E). The numbers of other inflammatory cell subsets, including CD11b$^+$F4/80$^+$ macrophages, CD11b$^+$CD11c$^+$ dendritic cells, and αβ TCR$^+$ T cells were not found to be significantly altered in *Il36r$^{fl/fl}$* skin in response to Aldara treatment at this time point (Fig S4).

Collectively, these data demonstrate the key instructive role that IL-36 signaling in keratinocytes plays in driving the recruitment to the skin of key pathogenic cell subsets required for the pathogenesis of psoriasiform inflammation. These observations add to the established importance of the IL-36 family in driving the pathogenesis of disease with implications for the future development of therapeutic strategies for patients. In particular, these data indicate that targeting of the IL-36 family specifically in keratinocytes, such as through the topical administration of specific inhibitors may provide a suitable approach. These observations also raise the possibility that IL-36 cytokines may play an important bridging mechanism between environmental factors, such as the skin microbiome and the pathogenesis of psoriatic inflammation. Along these lines, we, and others, have recently shown that these cytokines can alter the composition of the intestinal microbiome (Ngo et al, 2018; Giannoudaki et al, 2019). As this study demonstrates the significance of IL-36 activity on the outermost cells of the skin barrier in driving psoriatic inflammation, it will be of interest to determine whether IL-36 family members can play a similar instructive role on the skin microbiome.

## Materials and Methods

### Mice

All experiments were performed with 8–16-wk-old male and female mice. All animal experiments were performed with ethical approval by Trinity College Dublin Animal Research Ethics Committee and under license by the Irish Health Products Regulatory Authority (project authorization no: AE19136/P036).

### Generation of *Il36rΔK* mice

*Il1rl2floxed* (*Il36r$^{fl}$*) transgenic mice were generated on a C57Bl/6 background by Cyagen Biosciences (see Fig S1A for outline of strategy). Exon 4 was selected as the conditional KO region and deletion of this exon was predicted to result in loss of function of the mouse *Il1rl2* gene. Mouse genomic fragments containing homology arms and conditional KO region were amplified from the BAC clone by using high-fidelity Taq and were sequentially assembled into a targeting vector together with recombination sites and selection markers (Fig S1A). After confirming correctly targeted ES clones via Southern blotting, the clones were selected for blastocyst microinjection, followed by chimera production. Founders were confirmed as germ line–transmitted via crossbreeding with wild-type C57Bl/6 mice. Mutant mice received were heterozygous for the transgene. Heterozygous mice were mated to obtain homozygous in house (Fig S1B). Transgenic mice were identified by DNA extraction of ear tissue and amplification by PCR of the transgene. The K14$^{Cre}$ mice (B6N.Cg-Tg(KRT14-cre)1Amc/J) were obtained from Jackson Laboratories (Strain 004782). In these mice, the expression of Cre recombinase in keratinocytes is controlled by a human keratin 14 promoter. Il36rflox mice were mated with K14$^{Cre}$ mice to generate K14$^{Cre+}$Il36r$^{fl/fl}$ (*Il36rΔK*).

### Aldara model of psoriasiform inflammation

*Il36r$^{fl/fl}$*, *Il36rΔK*, and *Il36r$^{-/-}$* mice on C57BL/6 background were bred in house. *Il36r$^{-/-}$* mice were obtained from Amgen under Material Transfer Agreement that is described previously (Russell et al, 2016). Aldara cream (5% imiquimod; MEDA Pharmaceuticals) or Vaseline was applied to adjacent mice ears daily for up to 7 d. Ear thickness

analyses performed were one-way ANOVA multiple comparisons with Tukey's correction in Fig 3G–I, Mann–Whitney test used in Fig 3 A–C and E, and *t* test in Fig 3D. Significant differences are indicated as follows: *$P$ < 0.05; ***$P$ < 0.001; ****$P$ < 0.0001. Source data are available for this figure.

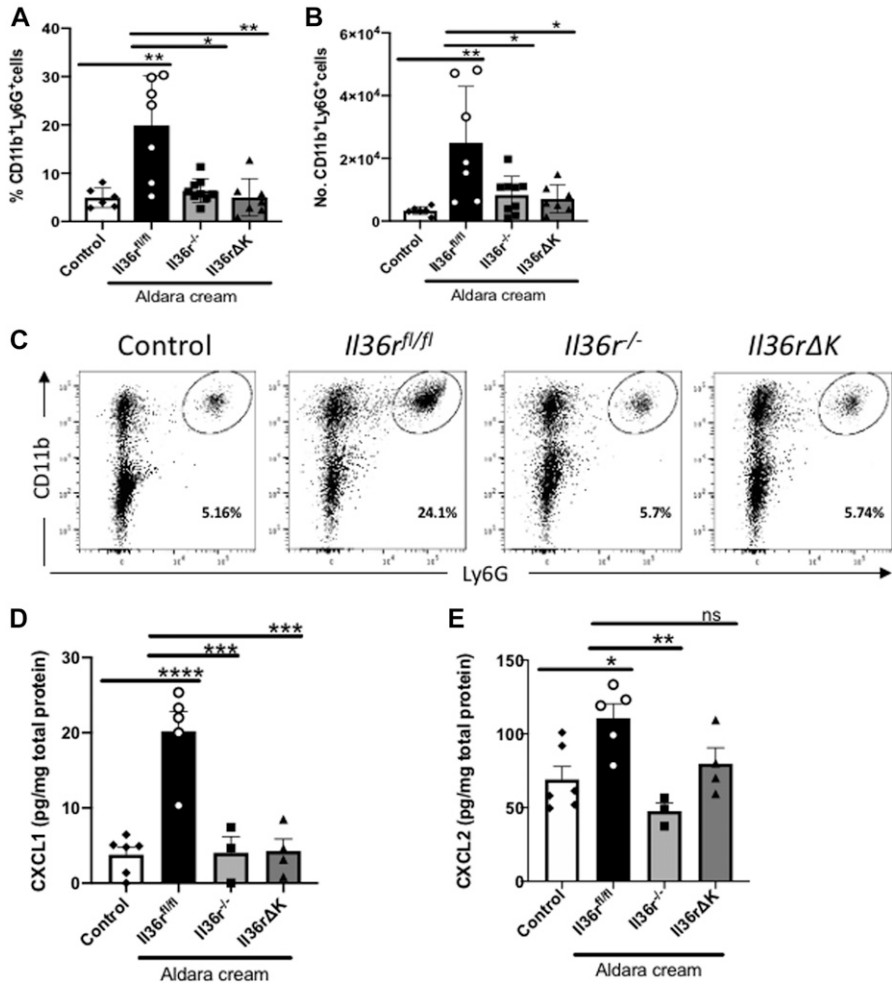

**Figure 4. Keratinocyte expression of *Il36r* directs neutrophil infiltration to the inflamed dermis.**
**(A, B)** Percentage and (B) total cell numbers of CD11b⁺Ly6G⁺ cells in the CD45⁺ cell compartment, from vehicle-treated littermate control (*n* = 6) as well as *Il36r*^fl/fl^ (*n* = 7), *Il36r*^−/−^ (*n* = 9), and *Il36rΔK* (*n* = 7) mice after 4-d Aldara treatment. **(C)** Representative plots for staining of CD11b⁺Ly6G⁺ neutrophils in vehicle-treated and the inflamed dermis of *Il36r*^fl/fl^, *Il36r*^−/−^, and *Il36rΔK* mice by flow cytometry. **(D, E)** Protein levels of Cxcl1 and Cxcl2 expressed as pg/mg of total protein from littermate control (*n* = 6), *Il36r*^fl/fl^ (*n* = 5), *Il36r*^−/−^ (*n* = 3), and *Il36rΔK* (*n* = 4) mice after 4-d Aldara treatment. Data show means ± SEM. Statistical analyses were performed using one-way ANOVA multiple comparisons with Tukey's correction except in Fig 4A and B in which Mann–Whitney test was used. Significant differences are indicated as follows: *$P <$ 0.05; **$P <$ 0.01; ***$P <$ 0.001; ****$P <$ 0.0001. Source data are available for this figure.

was measured before the initiation of the experiment (day 0) and every day subsequently on both Vaseline and Aldara-treated ears using a thickness gauge (Hitec).

### Histopathological analysis of tissue

Ear tissues were recovered and fixed overnight in 10% neutral buffered formalin (Medical Supply) before dehydration and embedding into paraffin blocks. Sections of 5 μm thickness were cut, stained with hematoxylin and eosin and scored blindly for pathological manifestations of psoriatic inflammation on a scale up to four (0, no differences over control; 1, mild; 2, moderate; 3, marked; 4, severe) for acanthosis, desquamation, parakeratosis, and infiltration. The scores for each parameter were combined into a total histological severity score.

### DAB staining

Ear tissue was recovered and frozen in OCT. 5-μm-thickness sections were obtained using a cryostat (Leica) and 3,3-DAB immunohistochemical staining was performed on the sections using

ImmPRESS Anti-Rag Ig Kit and ImmPACT DAB (Brown) peroxidase substrate (Vector Laboratories) following the manufacturer's instructions. The primary antibody used was mouse anti-IL-36R (M616; Amgen) at a dilution of 1/900. The sections were counterstained with Mayer's Hematoxylin for 5 min.

### ELISA

Ear tissues from *Il36r*^fl/fl^, *Il36rΔK*, and *Il36r*^−/−^ mice were lysed and homogenized with RIPA buffer (Sigma-Aldrich) containing protease and phosphatase inhibitors (Sigma-Aldrich) using Bead-Bug prefilled tubes with 1.5-mm Zirconium beads (Sigma-Aldrich) in a FastPrep-24 5G system (MP Biomedicals). Protein content of the lysates was quantified using Bicinchoninic Acid kit (Sigma-Aldrich) following the manufacturer's instructions. Levels of mouse cytokines Il-17A (homodimer), Il-23 (eBioscience), Cxcl1, and Cxcl2 (DuoSet ELISA; R&D Systems) were measured in protein lysates by ELISA following the manufacturer's instructions. Ears from *Il36r*^fl/fl^ and *Il36rΔK* littermates were used as non-inflamed control tissue to determine relative levels of induced protein expression.

### qRT-PCR

Ear tissues from mice were stored in RNAlater (Sigma-Aldrich) at −80°C. Isolate II RNA Mini kit (Bioline) was used to obtain total RNA following the manufacturer's instructions. The tissues were covered by lysis buffer and disrupted using BeadBug-prefilled tubes with 1.5-mm Zirconium beads (Sigma-Aldrich) in a FastPrep-24 5G system (MP Biomedicals). High-Capacity cDNA kit (Applied Biosystems) was used to perform reverse transcription. Real-time PCR was performed in triplicate using specific TaqMan Gene Expression Assays (Table 1) and TaqMan Fast Universal PCR Master Mix in a QuantStudio 3 System (Applied Biosystems). Normalization was performed using 18S ribosomal RNA, and relative gene expression levels were obtained by the ΔΔCt method. Levels of induced gene expression were determined by comparison with baseline levels found in the ear skin from *Il36r*$^{fl/fl}$ and *Il36rΔK* littermates used as non-inflamed control tissues.

### Keratinocytes and fibroblasts isolation

Keratinocytes and dermal fibroblast cells from *Il36r*$^{fl/fl}$, *Il36rΔK*, and *Il36r*$^{−/−}$ adult mouse skin were isolated using protocols described previously (Khan & Gasser, 2016; Li et al, 2017).

To assess the effects of IL-36 on both keratinocytes and fibroblasts, the cells were stimulated with recombinant mouse IL-36α (aa8-160) (R&D Systems) for 24 h, followed by analysis of gene expression for indicated genes by qRT-PCR.

### BMDC differentiation and stimulation

Bone marrow was extracted from the femur and tibia of *Il36r*$^{fl/fl}$, *Il36r*$^{−/−}$, and *Il36r*$^{ΔK}$ mice and cultured in 10 ml of complete RPMI (10% FCS + 1% penicillin streptomycin) (cRPMI) supplemented with 20 ng/ml recombinant mouse GM-CSF protein (Sigma-Aldrich) at 37°C. On days 3 and 6 of culture, the medium was refreshed with 10 ml of cRPMI + 20 ng/ml GM-CSF. On day 7, the cells were examined

**Table 1. TaqMan assays used in qRT-PCR to obtain gene expression data.**

| Gen | TaqMan assay code |
| --- | --- |
| Ccl20 | Mm01268754_m1 |
| Il17c | Mm00521397 |
| Il1f6/Il36a | Mm00457645_m1 |
| Il1f8/Il36b | Mm01337546_g1 |
| Il1f9/Il36g | Mm00463327_m1 |
| Il1rl2/Il36r | Mm00519245_m1 |
| Il22 | Mm01226722_g1 |
| Il23 | Mm01160011_g1 |
| l17a | Mm00439618_m1 |
| S100a8 | Mm00496696_g1 |
| S100a9 | Mm00656925_m1 |
| Cxcl1 | Mm04207460_m1 |

for CD11b/CD11c coexpression by flow cytometry and were then placed in culture at 3 × 10$^5$ cells/well with or without 100 ng/ml rmIL-36α (R&D Systems) for 24 h. After 24 h, the supernatants were harvested for analysis of Cxcl1 expression by ELISA (R&D Systems).

### Cell isolation for flow cytometry analysis

Ears were cut into small pieces that were digested with 3 mg/ml Dispase II in HBSS (Sigma-Aldrich) solution for 90 min at 37°C with agitation. After removing Dispase II solution, the tissues were digested with 1.5 mg/ml Collagenase D (Roche) in PBS for additional 90 min at 37°C with agitation. After digestion, the cell extracts were washed with RPMI with 10% FCS and 5 mM EDTA and disaggregated into single-cell suspension by dissociation of tissues and passing through 100- and 40-μm cell strainers consecutively.

### Flow cytometry

Surface and intracellular protein expression of cells was analysed using a BD LSRFortessa Cell Analyzer (BD Biosciences) with further analysis carried out using FlowJo software (BD Biosciences). Harvested cells from individual mice were first counted by trypan blue exclusion and an aliquot taken for analysis of CD45 expression levels to evaluate infiltrating cell numbers. For intracellular staining of IL-17a, the remaining cells were stimulated with PMA (10 ng/ml), ionomycin (1 μg/ml), and brefeldin A (5 μg/ml) during 4 h, then surface staining was performed and cells were fixed and permeabilized using eBioscience Foxp3/Transcription Factor Staining Buffer Set (eBiosciences) following the manufacturer's instructions. Alternatively, for analysis of neutrophils, the cells were stained immediately with the indicated surface markers, without restimulation and subsequently analysed. The following mouse antibodies were used for surface staining CD11c (N418), Ly6G (1A8), F4/80 (BM8), TCRβ chain (H57-597), CD45 (30-F11), CD11b, Vγ4 (UC3-10A6) (BioLegend), TCR gamma/delta (eBioGL3), and CD3e (145/2C11). Intracellular staining was performed using the mouse antibody Il-17a (eBio17B7). Dead cells were excluded from all analyses using Aqua Live/Dead stain (Invitrogen). All antibodies were purchased from eBioscience unless otherwise stated. The gating strategy used to identify and quantify immune cell subsets is shown in Fig S5.

### Statistical data analysis

Data were assessed for normal distribution and homoscedasticity. One-way ANOVA or Mann–Whitney U-test was applied as indicated in figure legends to compare the differences among the groups using Prism 8 software (GraphPad). Statistical significance details for each graph are indicated in the respective figure legends.

# Supplementary Information

## Acknowledgements

This work was supported by grant funding from the National Children's Research Centre to PT Walsh and an Irish Research Council postgraduate fellowship to G Leon.

## Author Contributions

YE Hernandez-Santana: conceptualization, data curation, formal analysis, investigation, and writing—original draft.
G Leon: data curation, formal analysis, investigation, and writing—original draft.
D St Leger: data curation and investigation.
PG Fallon: resources.
PT Walsh: conceptualization, data curation, formal analysis, investigation, and writing—original draft.

## Conflict of Interest Statement

The authors declare that they have no conflict of interest.

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
