## [Reviewer comments · Life Science Alliance]

Life Science Alliance

Keratinocyte Interleukin-36 receptor expression orchestrates psoriasiform inflammation in mice.

Yasmina Hernandez Santana, Gemma Leon, David St.Leger, Padraic Fallon, and Patrick Walsh
DOI: <https://doi.org/10.26508/lsa.201900586>

Corresponding author(s): Patrick Walsh, Trinity College Dublin

Review Timeline:	Submission Date:	2019-10-25
	Editorial Decision:	2019-10-28
	Revision Received:	2020-02-10
	Editorial Decision:	2020-02-12
	Revision Received:	2020-02-12
	Accepted:	2020-02-13

Scientific Editor: Andrea Leibfried

Transaction Report:

Please note that the manuscript was previously reviewed at another journal and the reports were taken into account in the decision-making process at Life Science Alliance.

Referee #1 Review

Report for Author:

The manuscript by Hernández-Santana et al describes the generation of IL-36R fl/fl mice, as a resource to study tissue/cell type-specific effects of IL-36 cytokines. These IL-1 family members have been shown to play a crucial role in the pathogenesis of psoriasis, both in mouse models and in humans. Aiming to determine the key target of IL-36 cytokines in the development of psoriasis, the authors bred IL-36R fl/fl animals with K14-Cre mice to specifically ablate IL-36R in keratinocytes (Il36rDeltaK) and prove that loss of IL-36R in keratinocytes confers resistance to IMQ-induced psoriasis, comparable to global IL-36R KO.

The novelty, in this respect, is relatively limited, as previous reports provided already strong hints for a key role of IL-36R expression on keratinocytes in this model of psoriasis. More specifically, as the authors mention in their manuscript, it had been previously shown that expression of IL-36R on radioresistant skin cells was required and sufficient to drive IMQ-induced psoriasis (Tortola et al, JCI 2012) and IL-36 treatment of keratinocytes was shown to drive upregulation of several key genes associated with the pathogenesis of psoriasis (Carrier et al, JID 2011; Hashiguchi et al, J Immunol 2018; Müller et al, PNAS 2018). Besides this, there are some points that should be addressed.

Major points:

1) To ensure the validity of the main point of the paper, i.e. that deletion of IL-36R on keratinocytes alone is enough to protect mice from IMQ-induced psoriasis, the authors need to prove that keratinocyte-specific deletion of IL-36R in their conditional knockouts is not leaky and therefore need to assess whether cells other than keratinocytes still express IL-36R and are activated by IL-36 ligands. Ideally, for this test the authors should compare keratinocytes to DCs and dermal fibroblasts (or CD45-negative cells from the dermis).

2) Figure 2: The authors state that loss of IL-36R in keratinocytes impairs IMQ-driven upregulation of different inflammatory mediators of psoriasis. Yet, it is unclear why the authors only show the significance of the statistical comparison between untreated control and *Il36rf1/fl* mice and do not show the results of the comparison between *Il36rf1/fl* and *Il36rDeltaK*, since this would be the relevant comparison for their statement. This should be rectified. A similar comment applies to the results shown in Figure 3 and 4.

3) In figure 3, the authors quantify the infiltration of neutrophils to the skin of IMQ-treated mice. The percentage of neutrophils among CD45+ cells is unusually and low compared to previous studies using the IMQ psoriasis model. Neutrophils have been reported to constitute a major fraction of the cells infiltrating the skin upon treatment with IMQ, and the percentages shown here are about 1 order of magnitude lower. According to the gating strategy shown in figure EV3, it appears as if the authors might have assessed the number of neutrophils from the same samples that were restimulated in vitro in order to quantify intracellular cytokine expression. Although they don't mention in the methods section how they performed restimulation, it can be assumed that they used PMA and ionomycin. While stimulating gene expression in T cells, PMA is also well known to drive NETosis and kill neutrophils, which would explain the unusually low numbers of neutrophils depicted in figure 3. The authors should therefore characterize the nature of skin-infiltrating cells in more detail, including major cell types (neutrophils, monocytes, ab/gd T cells, DCs, macrophages), and make sure to do so without previously treating isolated cells with PMA and ionomycin. Also, they should describe in better detail the procedure for intracellular cytokine staining in the methods section.

Minor point:

The title of the manuscript ("Expression of the Interleukin-36 receptor on keratinocytes is sufficient to orchestrate psoriasiform inflammation in mice") is incorrect: "required" and "sufficient" are not interchangeable. The authors prove that IL-36R expression in keratinocytes is essential for the development of IMQ-induced psoriasis. But to prove that IL-36R expression on keratinocytes is "sufficient" to drive psoriasis, they would have to knock-out IL-36R everywhere else and maintain expression exclusively on keratinocytes, and assess whether such mice develop psoriasis. The data presented here does not exclude the possibility that expression of IL-36R on both keratinocytes AND on an additional cell type (e.g. fibroblasts) is required for the development of psoriasiform dermatitis.

To circumvent this semantic issue, the authors could rephrase the title saying that The ABLATION of IL-36R in keratinocytes is sufficient to fully protect mice from IMQ-induced psoriasis (or something along those lines).

Referee #2 Review

Report for Author:

Comments on "Expression of the Interleukin-36 receptor on keratinocytes is sufficient to orchestrate psoriasiform inflammation in mice" by Dr. Hernandez-Santana and colleagues. In this manuscript the authors demonstrate that keratinocyte specific expression of Il36r is required to orchestrate psoriasiform disease. I have the following comments

There are already large amounts of data on the role of IL-36R in psoriasisform dermatitis and several papers have already tackled this and the contribution of different cellular subsets. IL-36R is primarily expressed on epithelial tissues and Tortola and colleagues showed that expression of IL-36R on radioresistant skin-resident cells, but not on hematopoietic bone marrow-derived cells, was essential for skin inflammation upon imiquimod (aldara) treatment. Although the data presented here show that it is the IL-36R on keratinocytes that are critical for driving IMQ skin inflammation, it represents an incremental advancement beyond what is already known.

Of note the IMQ (Aldara) mouse model is an acute form of skin inflammation. While it reflects some aspects of psoriasis it doesn't capture well the chronicity of this model. Based on the data provided here the authors show that IL-17a expression is diminished in the global IL-36R KO mice, whereas no difference in the frequency of the IL-17a producing T-cell subset was seen in the IL-36 epidermis knockout. This could suggest that IL-36 has a wider role beyond keratinocytes and that this is not captured in an acute model of psoriasis-like inflammation, particularly as this model of psoriasis only captures acute events and doesn't really address the contribution of IL-36 to activation of the adaptive immune system. The authors should consider adding data from a more chronic models of psoriasis but several such models already exist.

Page 5. "This model has previously been reported to be sensitive to treatment with frontline current therapeutic approaches currently in use in psoriasis patients [12,13,19]". I'm not sure that this is entirely true. The 1st and 2nd paper uses monoclonal antibody against the IL-36R in the IMQ mouse model. The third paper is practically a long list of anti-interleukin therapies for psoriasis and have nothing to do with the IMQ mouse model. To my knowledge none of the commonly used anti-psoriatic treatments have been used (or actually work) in the IMQ mouse model. Therefore, the authors should revise their statement. In fact, the use (and overuse) of this model has recently been questioned (see Hawkes JE et al, JID 2017).

Minor point

Page 6. " Similarly, treatment with anti-IL-36 receptor blocking antibodies can effectively suppress psoriasiform inflammation in this model". Please provide a reference for this.

Referee #3 Review

Report for Author:

The authors demonstrated that targeted deletion of Il36r in keratinocytes results in similar levels of protection from psoriasiform inflammation observed in 'global' Il36r deficient mice. The data demonstrated the key role that IL-36 signaling in keratinocytes plays in initiating psoriasiform inflammation. It is well-written and data are reliable. I think this paper add very important information on IL-36 studies on inflammatory skin diseases.

October 28, 2019

Re: Life Science Alliance manuscript #LSA-2019-00586-T

Dr. Patrick T Walsh
Trinity College Dublin
Clinical Medicine
School of Biochemistry and Immunology
Trinity Biomedical Sciences Institute, Trinity College Dublin
Dublin, Dublin 12
Ireland

Dear Dr. Walsh,

Thank you for transferring your manuscript entitled "Expression of the Interleukin-36 receptor on keratinocytes is sufficient to orchestrate psoriasiform inflammation in mice" to Life Science Alliance. The manuscript was assessed by expert reviewers at another journal before, and the editors transferred those reports to us with your permission.

The reviewers who evaluated your manuscript elsewhere appreciated your work, but would have expected a further reaching conceptual advance. This is not a concern for publication at Life Science Alliance, and we would thus like to invite you to submit a revised version of your manuscript to us.

We would expect a full point-by-point response and accordingly changes to the manuscript text/data representation. The experimental revision points requested by reviewer #1 should furthermore get addressed.

Thank you for this interesting contribution to Life Science Alliance. We are looking forward to receiving your revised manuscript.

Sincerely,

Andrea Leibfried, PhD
Executive Editor
Life Science Alliance
Meyerhofstr. 1

69117 Heidelberg, Germany
t +49 6221 8891 502
e a.leibfried@life-science-alliance.org
www.life-science-alliance.org

B. MANUSCRIPT ORGANIZATION AND FORMATTING:

Response to Reviewers:

Referee #1:

The manuscript by Hernández-Santana et al describes the generation of IL-36R fl/fl mice, as a resource to study tissue/cell type-specific effects of IL-36 cytokines. These IL-1 family members have been shown to play a crucial role in the pathogenesis of psoriasis, both in mouse models and in humans. Aiming to determine the key target of IL-36 cytokines in the development of psoriasis, the authors bred IL-36R fl/fl animals with K14-Cre mice to specifically ablate IL-36R in keratinocytes (Il36rDeltaK) and prove that loss of IL-36R in keratinocytes confers resistance to IMQ-induced psoriasis, comparable to global IL-36R KO.

The novelty, in this respect, is relatively limited, as previous reports provided already strong hints for a key role of IL-36R expression on keratinocytes in this model of psoriasis. More specifically, as the authors mention in their manuscript, it had been previously shown that expression of IL-36R on radioresistant skin cells was required and sufficient to drive IMQ-induced psoriasis (Tortola et al, JCI 2012) and IL-36 treatment of keratinocytes was shown to drive upregulation of several key genes associated with the pathogenesis of psoriasis (Carrier et al, JID 2011; Hashiguchi et al, J Immunol 2018; Müller et al, PNAS 2018). Besides this, there are some points that should be addressed.

Major points:

1) To ensure the validity of the main point of the paper, i.e. that deletion of IL-36R on keratinocytes alone is enough to protect mice from IMQ-induced psoriasis, the authors need to prove that keratinocyte-specific deletion of IL-36R in their conditional knockouts is not leaky and therefore need to assess whether cells other than keratinocytes still express IL-36R and are activated by IL-36 ligands. Ideally, for this test the authors should compare keratinocytes to DCs and dermal fibroblasts (or CD45-negative cells from the dermis).

Response: We thank the referee for this helpful comment. As suggested, we have now examined IL-36 induced responses of keratinocytes, dermal fibroblasts and bone marrow derived dendritic cells from the *Il36rΔK* mice. These new data have confirmed that while responsiveness is lost in *Il36rΔK* keratinocytes (Figure 1C), it is maintained in both dermal fibroblasts (Figure 1D) and DCs (sFigure 3) isolated from these mice. Importantly, loss of IL-36 responsiveness is found in all cell types examined from the *Il-36r*^{-/-} mouse. We believe this new data indicates that in our model loss of the *il36r* responsiveness is restricted to keratinocyte cells.

2) Figure 2: The authors state that loss of IL-36R in keratinocytes impairs IMQ-driven upregulation of different inflammatory mediators of psoriasis. Yet, it is unclear why the authors only show the significance of the statistical comparison between untreated control and Il36rfl/fl mice and do not show the results of the comparison between Il36rfl/fl and Il36rDeltaK, since this would be the relevant comparison for their statement. This should be rectified. A similar comment applies to the results shown in Figure 3 and 4.

Response: We have now included a more complete statistical analysis including the comparisons suggested in Figures 2,3 & 4.

3) In figure 3, the authors quantify the infiltration of neutrophils to the skin of IMQ-treated mice. The percentage of neutrophils among CD45+ cells is unusually and low compared to previous studies using the IMQ psoriasis model. Neutrophils have been reported to constitute a major fraction of the cells infiltrating the skin upon treatment with IMQ, and the percentages shown here are about 1 order of magnitude lower. According to the gating strategy shown in figure EV3, it appears as if the authors might have assessed the number of neutrophils from the same samples that were restimulated in vitro in order to quantify

intracellular cytokine expression. Although they don't mention in the methods section how they performed restimulation, it can be assumed that they used PMA and ionomycin. While stimulating gene expression in T cells, PMA is also well known to drive NETosis and kill neutrophils, which would explain the unusually low numbers of neutrophils depicted in figure 3. The authors should therefore characterize the nature of skin-infiltrating cells in more detail, including major cell types (neutrophils, monocytes, ab/gd T cells, DCs, macrophages), and make sure to do so without previously treating isolated cells with PMA and ionomycin. Also, they should describe in better detail the procedure for intracellular cytokine staining in the methods section.

Response: We thank the referee for this helpful critique. In figure 3 of the original submission, we did analyse neutrophils in the same samples that were subject to restimulation in vitro with PMA and Ionomycin which likely resulted in the relatively low numbers detected. We apologise for the lack of clarity surrounding this issue in our original Methods section.

To address this, we repeated these experiments to separately analyse the infiltration and activation of both neutrophils and $\gamma\delta$ T cells at day 4 post Aldara administration, to allow analysis of neutrophil infiltration in the absence of re-stimulation. As suggested, these data, now shown in Figure 4, demonstrate a higher magnitude of neutrophil infiltration in the skin of control mice in agreement with previous reports. We also reanalysed the cellular expression of IL-17A at day 4, given our findings of reduced levels of IL-17A protein expression in the skin at this timepoint as determined by ELISA, and the acute nature of this model. This new data, now shown in Figure 3 of the revised manuscript, demonstrates that both the skin infiltrating numbers and expression levels of IL-17a by $V\gamma 4+$ $\gamma\delta$ T cells are reduced in *I36r Δ K* mice. We believe this new data, generated with greater experimental numbers demonstrates a more robust reflection of how these important inflammatory cell responses are diminished in the absence of keratinocyte specific IL-36r expression. We believe these new data substantially improve the overall manuscript and we have clarified our experimental approach in the methods section as suggested. We have also examined the relative numbers of $\alpha\beta$ T cells, macrophages and DC in the inflamed skin of these mice and this data is now included as supplementary figure 4.

Minor point:

The title of the manuscript ("Expression of the Interleukin-36 receptor on keratinocytes is sufficient to orchestrate psoriasiform inflammation in mice") is incorrect: "required" and "sufficient" are not interchangeable. The authors prove that IL-36R expression in keratinocytes is essential for the development of IMQ-induced psoriasis. But to prove that IL-36R expression on keratinocytes is "sufficient" to drive psoriasis, they would have to knock-out IL-36R everywhere else and maintain expression exclusively on keratinocytes, and assess whether such mice develop psoriasis. The data presented here does not exclude the possibility that expression of IL-36R on both keratinocytes AND on an additional cell type (e.g. fibroblasts) is required for the development of psoriasiform dermatitis.

To circumvent this semantic issue, the authors could rephrase the title saying that The ABLATION of IL-36R in keratinocytes is sufficient to fully protect mice from IMQ-induced psoriasis (or something along those lines).

Response: We have changed the title of our manuscript as suggested to more accurately reflect our findings and meet the character limits set by the journal.

Referee #2:

Comments on "Expression of the Interleukin-36 receptor on keratinocytes is sufficient to orchestrate psoriasiform inflammation in mice" by Dr. Hernandez-Santana and colleagues. In this manuscript the authors

demonstrate that keratinocyte specific expression of IL36r is required to orchestrate psoriasiform disease. I have the following comments

There are already large amounts of data on the role of IL-36R in psoriasiform dermatitis and several papers have already tackled this and the contribution of different cellular subsets. IL-36R is primarily expressed on epithelial tissues and Tortola and colleagues showed that expression of IL-36R on radioresistant skin-resident cells, but not on hematopoietic bone marrow-derived cells, was essential for skin inflammation upon imiquimod (aldara) treatment. Although the data presented here show that it is the IL-36R on keratinocytes that are critical for driving IMQ skin inflammation, it represents an incremental advancement beyond what is already known.

Of note the IMQ (Aldara) mouse model is an acute form of skin inflammation. While it reflects some aspects of psoriasis it doesn't capture well the chronicity of this model. Based on the data provided here the authors show that IL-17a expression is diminished in the global IL-36R KO mice, whereas no difference in the frequency of the IL-17a producing T-cell subset was seen in the IL-36 epidermis knockout. This could suggest that IL-36 has a wider role beyond keratinocytes and that this is not captured in an acute model of psoriasis-like inflammation, particularly as this model of psoriasis only captures acute events and doesn't really address the contribution of IL-36 to activation of the adaptive immune system. The authors should consider adding data from a more chronic models of psoriasis but several such models already exist.

Response: We agree with the referees comment that analysis of our findings in a more chronic model of psoriasis would significantly enhance our data. However, such an analysis is unfortunately beyond the scope of this brief report.

Page 5. "This model has previously been reported to be sensitive to treatment with frontline current therapeutic approaches currently in use in psoriasis patients [12,13,19]". I'm not sure that this is entirely true. The 1st and 2nd paper uses monoclonal antibody against the IL-36R in the IMQ mouse model. The third paper is practically a long list of anti-interleukin therapies for psoriasis and have nothing to do with the IMQ mouse model. To my knowledge none of the commonly used anti-psoriatic treatments have been used (or actually work) in the IMQ mouse model. Therefore, the authors should revise their statement. In fact, the use (and overuse) of this model has recently been questioned (see Hawkes JE et al, JID 2017).

Response: We acknowledge the short comings of the IMQ mouse model as a reflection of human psoriasis. Similar to many murine models of inflammatory disease, this can, for the most part, be ascribed to its acute nature as well as some notable differences in inflammatory cell subsets observed between mouse and human skin. Notwithstanding these issues, the IMQ model has been extensively shown to be responsive to anti-psoriatic treatments used in patients including, anti-TNF, anti-IL-23/12 and anti-IL-17 neutralisation mAbs (Pantelyushin et al., J.Clin.Invest. 2012;122(6):2252, Shibata et al. Nat.Comm. 2015; 6, 7687, Mitsui et al. ClinImmunol. 2015;157 (1); 43, Shimizu et al., J.Derm.Sci. 2019: 95; 90), UVB light exposure (Racz et al. PlosOne. 2011; 6(5) e19806) and topical interventions (Sun et al. Immunopharmacol.Immunotoxicol. 2014; 36 (1) 17, Gabriel et al. J.Controlled Release, 2016; 242.16). We apologise for not including the appropriate references to illustrate this point in the original manuscript and have now corrected this.

Minor point

Page 6. " Similarly, treatment with anti-IL-36 receptor blocking antibodies can effectively suppress psoriasiform inflammation in this model". Please provide a reference for this.

Response: We have now included a relevant reference as suggested.

Referee #3:

The authors demonstrated that targeted deletion of IL36r in keratinocytes results in similar levels of protection from psoriasiform inflammation observed in 'global' IL36r deficient mice. The data demonstrated the key role that IL-36 signaling in keratinocytes plays in initiating psoriasiform inflammation. It is well-written and data are reliable. I think this paper add very important information on IL-36 studies on inflammatory skin diseases.

February 12, 2020

RE: Life Science Alliance Manuscript #LSA-2019-00586-TR

Dr. Patrick T Walsh
Trinity College Dublin
Clinical Medicine
School of Biochemistry and Immunology
Trinity Biomedical Sciences Institute, Trinity College Dublin
Dublin, Dublin 12
Ireland

Dear Dr. Walsh,

Thank you for submitting your revised manuscript entitled "Keratinocyte Interleukin-36 receptor expression orchestrates psoriasiform inflammation in mice". I now assessed the data added in revision and your point-by-point response and I appreciate the introduced changes. I would thus be happy to publish your paper in Life Science Alliance pending final minor revisions:

- If I read your figures correctly, Aldara treatment occurred for 6 days for Fig. 4D and E - please add this information in the figure legend
- The uploaded figures (tif files) are not of production quality => please make sure that all text is readable (please replace figures S5, 3, 4 with better versions)
- Please make sure that the author order in our submission system is the same as in the word docx file provided
- Please list 10 authors et al in your reference list

A. FINAL FILES:

-- High-resolution figure, supplementary figure and video files uploaded as individual files: See our detailed guidelines for preparing your production-ready images, <http://www.life-science->

alliance.org/authors

B. MANUSCRIPT ORGANIZATION AND FORMATTING:

Sincerely,

Andrea Leibfried, PhD
Executive Editor
Life Science Alliance
Meyerhofstr. 1
69117 Heidelberg, Germany
t +49 6221 8891 502
e a.leibfried@life-science-alliance.org

February 13, 2020

RE: Life Science Alliance Manuscript #LSA-2019-00586-TRR

Dr. Patrick T Walsh
Trinity College Dublin
Clinical Medicine
School of Medicine
Trinity Translational Medicine Institute.
Dublin, Dublin 12
Ireland

Dear Dr. Walsh,

Thank you for submitting your Research Article entitled "Keratinocyte Interleukin-36 receptor expression orchestrates psoriasiform inflammation in mice.". It is a pleasure to let you know that your manuscript is now accepted for publication in Life Science Alliance. Congratulations on this interesting work.

DISTRIBUTION OF MATERIALS:

Again, congratulations on a very nice paper. I hope you found the review process to be constructive and are pleased with how the manuscript was handled editorially. We look forward to future exciting

submissions from your lab.

Sincerely,
